# Evaluating the impact of global fund withdrawal on needle and syringe provision, cost and use among people who inject drugs in Tijuana, Mexico: a costing analysis

Javier A Cepeda,[1] Jose Luis Burgos,[1] James G Kahn,[2] Rosario Padilla,[3] Pedro Emilio Meza Martinez,[4] Luis Alberto Segovia,[5] Tommi Gaines,[1] Daniela Abramovitz,[1] Gudelia Rangel,[6] Carlos Magis-Rodriguez,[7] Peter Vickerman,[8] Steffanie A Strathdee,[1] Natasha K Martin[1,8]

JAC and JLB contributed equally.

For numbered affiliations see end of article.

**Correspondence to**
Dr Javier A Cepeda;
jacepeda@ucsd.edu

## ABSTRACT

**Objective** From 2011 to 2013, the Global Fund (GF) supported needle and syringe programmes in Mexico to prevent transmission of HIV among people who inject drugs. It remains unclear how GF withdrawal affected the costs, quality and coverage of needle and syringe programme provision.

**Design** Costing study and longitudinal cohort study.

**Setting** Tijuana, Mexico.

**Participants** Personnel from a local needle and syringe programme (n=6) and people who inject drugs (n=734) participating in a longitudinal study.

**Primary outcome measures** Provision of needle and syringe programme services and cost (per contact and per syringe distributed, in 2017 $USD) during GF support (2012) and after withdrawal (2015/16). An additional outcome included needle and syringe programme utilisation from a concurrent cohort of people who inject drugs during and after GF withdrawal.

**Results** During the GF period, the needle and syringe programme distributed 55 920 syringes to 932 contacts (60 syringes/contact) across 14 geographical locations. After GF withdrew, the needle and syringe programme distributed 10 700 syringes to 2140 contacts (five syringes/contact) across three geographical locations. During the GF period, the cost per harm reduction contact was approximately 10-fold higher compared with after GF ($44.72 vs $3.81); however, the cost per syringe distributed was nearly equal ($0.75 vs $0.76) due to differences in syringes per contact and reductions in ancillary kit components. The mean log odds of accessing a needle and syringe programme in the post-GF period was significantly lower than during the GF period (p=0.02).

**Conclusions** Withdrawal of GF support for needle and syringe programme provision in Mexico was associated with a substantial drop in provision of sterile syringes, geographical coverage and recent clean syringe utilisation among people who inject drugs. Better planning is required to ensure harm reduction programme sustainability is at scale after donor withdrawal.

### Strengths and limitations of this study

► We analysed provision and cost data of a needle and syringe programme in Tijuana, Mexico during and after Global Fund withdrawal which we used to estimate how withdrawal impacted quality of the programme.

► Our findings were further strengthened with the triangulation of self-reported needle and syringe programme utilisation data from a concurrent cohort of people who inject drugs in Tijuana.

► We were uncertain about the number of unique clients of the needle and syringe programme since only the number of contacts (kits distributed) was provided.

## BACKGROUND

The effectiveness of needle and syringe programmes (NSP) in reducing transmission of HIV and hepatitis C virus (HCV) among people who inject drugs (PWID) has been well documented. Findings from a meta-analysis reported that NSPs from higher quality studies were associated with a 58% (95% CI 0.22 to 0.81) reduction in HIV transmission.[1] Similarly, a recent Cochrane systematic review and meta-analysis found that NSPs were associated with a 21% reduction in HCV transmission (RR=0.79, 95% CI 0.39 to 1.61), although a stronger effect was seen in Europe (RR=0.24, 95% CI 0.09 to 0.62).[2] Despite the protective benefits of these services, the coverage of critical harm reductions services such as NSPs remains suboptimal, especially in low/middle-income countries (LMIC)[3] where most of the HIV and HCV disease burden lies.[4]

Programme evaluation, such as costing analyses, is important for budgeting and can help policymakers make evidence-based decisions with scarce resources. While LMIC would benefit the most from costing analyses of harm reduction services due to these countries having more limited resources, few economic evaluations of harm reduction services have been published in these settings. Studies conducted in Eastern Europe,[5 6] Bangladesh[7] and China[8] showed that harm reduction services can be effective relative to their cost, especially within the context of nascent HIV epidemics among PWID. In Latin America, there have been no economic evaluations of NSPs.

Despite sharing one of the busiest land border crossings in the world, numerous socioeconomic and health disparities separate Tijuana, Mexico from San Diego, California. Tijuana has a prominent red-light district and draws in drug and sex tourists primarily from the USA that has resulted in a localised HIV epidemic.[9] It also has one of the highest concentrations of PWID in Mexico, 4%–10% of whom are HIV infected and >90% of whom are HCV antibody positive.[10 11] NSPs have been operating in Tijuana for more than 15 years; however, prevention of transmission remains a challenge. The proportion accessing harm reduction services (<10% in the last 6 months in 2011) is lower than the coverage recommended by the WHO[12] who defined 'good coverage' as >60% of PWID contacting NSP services at least monthly in the past year.[13]

From 2011 to 2013, the Global Fund (GF) supported NSP provision in Mexico. However, due to Mexico's rising gross domestic product, the GF abruptly withdrew support by December 2013. It is unclear how this withdrawal affected the provision and economics of NSPs in Mexico. Our analysis had two objectives: (1) to compare NSP operations and costs between two periods, in 2012 (when NSPs were receiving funding from the GF) and in 2015 (after GF stopped funding projects in Mexico); and (2) to examine the effect of GF withdrawal on NSP access from PWID enrolled in a longitudinal cohort study in Tijuana. Findings from this analysis may inform harm reduction provision planning and donor support planning in other settings across the region, particularly those who may transition from donor-funded to state-funded harm reduction provision.

## METHODS
Harm reduction provision and cost data were collected from one NSP site in Tijuana, Mexico. Data collection occurred between March 2016 and February 2017. We examined cost data on NSP provision and cost for two periods: during GF support (2012) and after the GF withdrew support (2015/2016). To estimate provision and costs of an efficient NSP with enhanced resources, we report outcomes during the highest volume month of GF support (May 2012). To estimate current provision and costs of NSP, we report average monthly outcomes for 2015/2016, and additionally report on provision during the highest volume month of 2015/2016 (July 2015) for direct comparison with May 2012.

### NSP characteristics
The NSP was a fixed site located in the Zona Centro (near the 'Red-Light' district, a hotspot of illicit drug use and commercial sex activity). Distribution of the number of syringes per contact was reliant on available funding; however, they were provided at no cost to the user. NSP operated 11 months per year, and provided sterile syringes, offered rapid HIV testing/counselling and referred to hepatitis B and C testing/counselling.

### Service provision data collection
During site visits, we reviewed daily logs of geographical outreach of needle and syringe activities, contents of sterile syringe kits and operating hours. We obtained estimates on the number of contacts and number of syringes distributed per month from activity logs provided by senior staff.

### Costing strategy
We costed from an economic perspective, monetising all input resources, including staff, supplies (purchased or donated), building space and other items. We used an ingredients-based top-down[14] microcosting approach where overall inputs were measured at the programmatic level (ie, we did not observe individual clients or services) separately and combined to generate total and per-client unit costs. We divided total monthly costs by 2 monthly outputs of interest: (1) number of harm reduction contacts and (2) number of sterile syringe kits distributed. The study evaluated current implementation costs and did not consider start-up costs since the NSP has been in operation by a non-governmental organisation for several years with support from the federal HIV/AIDS prevention agency in Mexico (Centro Nacional para la Prevención y el Control del VIH y el Sida (CENSIDA)).

We classified costs as recurrent (eg, personnel and non-personnel) and capital. Personnel salaries were obtained from expenditure records, and the number of hours and percent effort dedicated to operating the NSP (including administration) were obtained from interviews with senior staff. During the GF investment period, outreach workers were paid per harm reduction kit distributed. Volunteer costs were calculated based on interviews with senior staff that reported the number of NSP-related hours and the wage a volunteer would have received if they had been employed. Recurrent non-personnel costs included supplies (including syringes and ancillary harm reduction items), building maintenance, utilities and other services (accounting, maintenance, cleaning, security, etc). Unit prices for inputs were obtained from financial records, itemised bills/receipts and sales catalogues. Capital costs included building space and equipment. Senior staff provided an overall monthly rent and an estimate of the proportion of building space that was

attributed only to provision of NSP services, which we confirmed visually during site visits. We multiplied the rent by the proportion dedicated to NSP services to obtain the operational cost for only NSP provision. All recurrent costs associated with operating a vehicle (fuel, insurance, etc) were obtained from expenditure records. Equipment and vehicle costs were amortised over the estimated lifespan of the item and then converted into a monthly cost. During the GF period, fuel costs for May 2012 were estimated using daily transportation logs, which were used to calculate miles driven. We used this to convert to estimated litres consumed using estimates of the vehicle's fuel economy multiplied by fuel prices (1 L=US$0.78).[15]

### Costing period during GF

We obtained activity logs for May 2012. Since provision of services varied during the GF period, we intentionally selected this month because it reflected a period of maximum (ie, 'ideal') provision with GF investment according to interviews with study staff which we then contrasted with current levels of provision (post-GF). These personnel costs were inflated to 2017 Mexican pesos using the consumer price index from the Instituto Nacional de Estadística y Geografía and then converted to US dollars using the January 2017 exchange rate (20.72 Mexican pesos=US$1).

### Costing period after GF

In addition to cost data during the GF period, we also obtained data after GF withdrawal from 2015 to 2016 (costing period May–April). Costs were inflated to 2017 Mexican pesos using the midpoint of each cost year, and then converted to US dollars.

### NSP access among PWID living in Tijuana

Since 2011, members of our research division have followed a cohort of PWID (n=734) living in Tijuana (Project 'El Cuete IV') to assess trends and patterns in risk behaviours, HIV incidence and harm reduction service utilisation. Study procedures have been described elsewhere and all participants consented to study procedures.[11] To assess how GF withdrawal may have impacted clean syringe provision among PWID, we analysed data collected from March 2012 to June 2016 (roughly coinciding with the end of the costing period) to determine the proportion of El Cuete participants who reported receiving clean syringes from an NSP within the past 6 months. We applied methods from interrupted time series analysis and conducted segmented regression[16] to estimate significant temporal changes in NSP utilisation during the GF period and then after the GF withdrew. We first fit a logistic regression model with fixed and random effects and a first-order autoregressive correlation structure to generate the mean predicted probabilities for each quarter of the calendar year. We then fit the mean predicted probabilities into a segmented linear regression model controlling for autoregressive error to estimate the coefficients of accessing a NSP during the

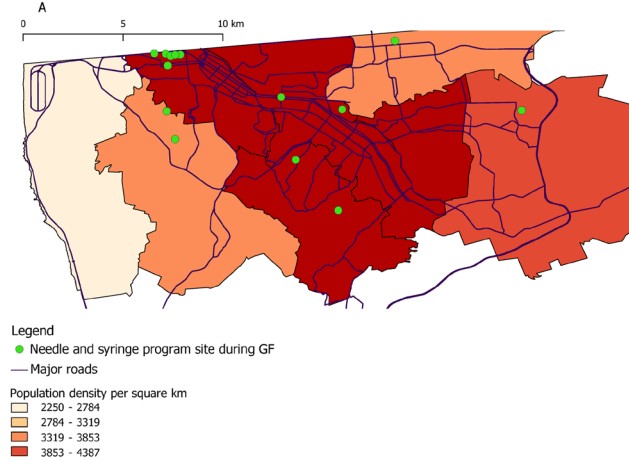

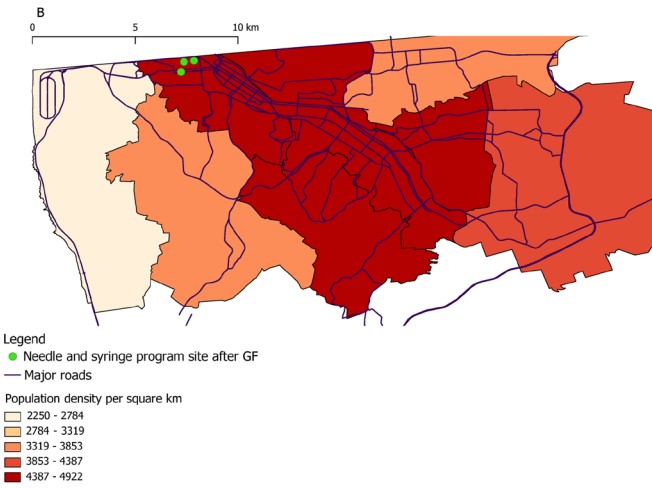

Source: GIS data publicly available from the San Diego Association of Governments and Tijuana Metropolitan Institute of Planning

**Figure 1** Geographical coverage of needle and syringe programme outreach sites (A) during Global Fund (GF) period in May 2012 and (B) after GF withdrawal in 2015/2016. Geographical information system (GIS) data of the major roads and administrative districts of Tijuana are publicly available from the San Diego Association of Governments (SANDAG)[31] and the Tijuana Metropolitan Institute of Planning, respectively.[32]

different GF periods. Additional details are provided in the online supplementary material.

### Patient and public involvement

We presented the study to community stakeholders and obtained their approval. Study staff, who had formerly used drugs and reflected the community, reviewed and pretested the survey.

## RESULTS
### NSP provision and utilisation during GF period (2012)

In 2012, the NSP provided harm reduction services 6 days per week and offered outreach services which covered a wide geographical area across Tijuana (figure 1A).

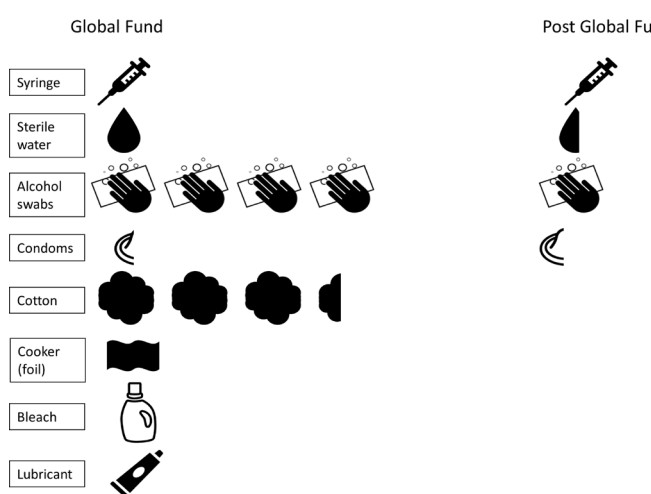

**Figure 2** Harm reduction kit components per sterile syringe distributed during the Global Fund (GF) period and after withdrawal of GF. Note: Some items are only partially depicted since more syringes per item were distributed (eg, during the GF period, three syringes were distributed per one condom).

Contents of the harm reduction kit included: 60 syringes (all low dead space), 240 alcohol swabs, 20 condoms, 60 3 mL vials of sterile water, 200 cotton swabs, one aluminium sheet, one 60 g tube of lubricant and one bottle of bleach (figure 2). In May 2012, the NSP reported 932 harm reduction contacts, resulting in 55 920 syringes distributed (table 1). Personnel employed during this time included: a coordinator, accountant, counsellor and nurse. Eight health education/outreach workers conducted needle and syringe at various sites and 115 HIV tests were conducted. Based on the monthly activity log, we estimated that 234 km per month were travelled for outreach purposes.

### Cost of NSP provision during GF period (2012)

The total monthly cost of NSP provision in May 2012 was $41 681 (table 2). The cost per harm reduction contact was $44.72 while the cost per syringe distributed was $0.75. Nearly two-thirds (62%) of the total monthly costs were attributed to ancillary kit components (shown in figure 2; $26,262) while syringes contributed 27% ($11,395) and 10% consisted of recurrent personnel, non-personnel and capital costs. The cost of syringe distribution excluding ancillary kit components (personnel+other recurrent+capital+syringes only) during the GF period was $0.28 per syringe distributed.

### NSP provision and utilisation post-GF (2015/2016)

Operations and provision of harm reduction services differed substantially during and after the GF withdrew support in 2015/2016 (figure 1B and table 1). As shown in figure 1B, geographic coverage of providing harm reduction services was sharply reduced after the GF withdrawal and limited mostly to the Zona Norte. Outreach personnel was reduced from eight workers in 2012 to four workers in 2015/2016. Additionally, post-GF, the harm reduction kit contents were substantially reduced compared with the GF period (table 1 and figure 2). In 2015, the harm reduction kit contents included five syringes (three low-dead space and two high-dead space, compared with 60 low-dead space syringes in 2012), three 3 mL vials of sterile water, five alcohol swabs and three condoms. Additionally, post-GF,

**Table 1** Provision of NSP services during the GF period (May 2012), highest volume (July 2015) and average month (2015/2016)

| | GF period | Post-GF period | |
| --- | --- | --- | --- |
| | **May 2012** | **2015/2016 (highest volume month)** | **Post-GF period 2015/2016 (average month)** |
| Unit of service | | | |
| Harm reduction contacts per month | 932 | 3170 | 2140 |
| Contents of harm reduction kit | | | |
| Syringes | 60 | 5 | 5 |
| Sterile water | 60 | 3 | 3 |
| Alcohol swabs | 240 | 5 | 5 |
| Cotton | 200 | – | – |
| Foil | 1 | – | – |
| Bleach | 1 | – | – |
| Condoms | 20 | 2 | 2 |
| Lubricant (60 g) | 1 | – | – |
| Syringes provided | 55920 | 15850 | 10700 |
| Number of HIV tests conducted | 115 | --* | 55 |

*HIV testing data were not available for the highest volume month.
GF, Global Fund; NSP, needle and syringe programmes.

**Table 2** Capacity, optimum (May 2012) and average monthly costs of NSPs operating in Tijuana, Mexico (all costs in 2017 $US)

| | GF period | Post GF period |
| --- | --- | --- |
| | May 2012 | 2015/2016 (average month) |
| Unit of service | | |
| Harm reduction contacts per month | 932 | 2140 |
| Capital cost (monthly) | $778 | $778 |
| Building/space* | $537 | $537 |
| Equipment | $241 | $241 |
| Personnel* (monthly) | $2503 | $2407 |
| Coordinators | $748 | $748 |
| Accountant | $94 | $94 |
| Counsellor/Head of harm reduction services | $424 | $424 |
| Clinician (Nurse/Physician) | $698 | $698 |
| Health educators/Outreach workers | $539 | $443 |
| Non-personnel recurrent costs (monthly) | $38399 | $4947 |
| Syringes | $11395 | $1853 |
| Supplies | $178 | $178 |
| Utilities and other services* | $564 | $485 |
| Ancillary harm reduction contents | $26262 | $2431 |
| Total monthly cost† | $41681 | $8131 |
| Cost per harm reduction contact | $44.72 | $3.80 |
| Cost per syringe distributed including ancillary kit contents | $0.75 | $0.76 |
| Cost per syringe distributed excluding ancillary kit contents | $0.28 | $0.53 |

*Only includes the amount dedicated to providing harm reduction services.
†May not sum to total due to rounding.
GF, Global Fund; NSP, needle and syringe programmes.

service provision was reduced by 1 day to 5 days per week. Despite decreased geographic coverage and reduced opening hours, the number of kits distributed per month was higher in 2015 than in 2012. NSP reported a mean of 2140 monthly contacts in 2015/2016 (3170 contacts during highest volume month of 2015/2016), compared with 932 in May 2012. However, because of the substantial decrease in syringes per kit, the total number of syringes distributed was substantially lower in 2015/2016—an average of 10 700 syringes distributed per month in 2015/2016. During the highest volume month in 2015/2016, there were 15 850 syringes distributed compared with 55 920 in May 2012.

### Cost of NSP provision post-GF (2015/2016)

The total monthly cost of the NSP was over fivefold higher during 2012 compared with the average monthly cost during 2015/2016 ($41 681 vs $8131, respectively), primarily due to the higher cost of the harm reduction kit during the GF period ($44.72 vs $3.80 per kit, respectively). By comparison, the total monthly cost during the month of maximum provision, post-GF, was $10 193. The cost of just syringe distribution (excluding ancillary kit components) doubled from $0.28/syringe distributed in 2012 to $0.53/syringe distributed in 2015/2016, mostly

due to the reduction in syringes distributed and thus the higher personnel cost per syringe (figure 3). However, after including ancillary kit components, there was no change in the cost per syringe distributed in 2015/2016 ($0.76) compared with May 2012 ($0.75) because of the

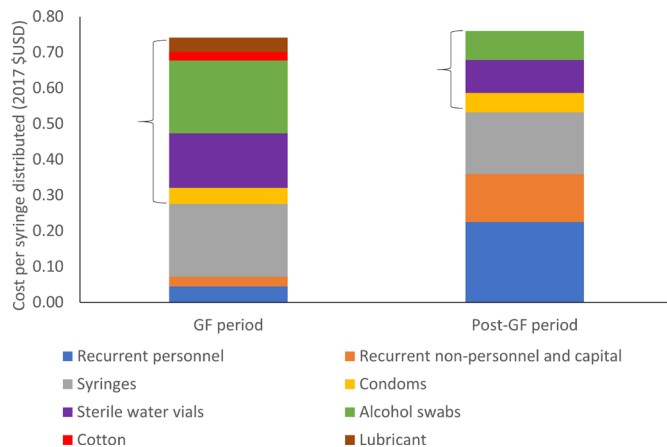

**Figure 3** Cost breakdown per syringe distributed at the needle and syringe programmes site (ancillary harm reduction components in brackets). GF, Global Fund.

reduction in ancillary kit component expenditure. Similarly, there was little change in cost per syringe ($0.64) distributed when comparing with the maximum volume month in 2015/2016. Harm reduction kit (syringes+ancillary components) comprised 90% of the costs per syringe distributed during the GF period, whereas these items comprised only 51% in 2015/2016 (figure 3).

### Temporal trends in NSP access among PWID

Based on data from PWID in the El Cuete IV study, we calculated the mean predicted probabilities of accessing NSP over the 17 3-month periods, which are shown in online supplementary figure S1. Overall, there was a significant increasing trend in the probability of accessing the NSP during the GF period, which peaked in September 2013 (51%, 95% CI 42% to 59%). During the GF period, the mean log odds of accessing NSP increased by a factor of 0.17 (p value<0.001). The immediate change that occurred between the end of the last 3 months of the GF period and the end of the first 3 months of the post-GF period was associated with a 0.73 reduction in the mean log odds of accessing NSP in the past 6 months (p=0.02). During the post-GF period, the mean log odds of accessing NSP decreased by a factor of 0.22 (p=0.002).

### DISCUSSION

Our analysis is among the first to describe the cost of providing needle and syringe services in a Latin American setting and the first to specifically compare coverage and costs of needle and syringe during versus after withdrawal of GF support. We found dramatic declines in geographical coverage and number of syringes and ancillary kit components distributed post-GF withdrawal among one NSP provider in Tijuana, with concomitant declines in reported syringe access among PWID. Excluding ancillary kit components, cost per syringe distributed doubled post-GF; total cost per syringe (including kit components) remained similar across periods as ancillary components were dramatically reduced to cut costs. We expect to use both GF ('ideal') and post-GF ('current') NSP provision costs to inform future cost-effectiveness analyses of NSPs on reducing HIV incidence in LMIC.

Despite dramatic declines in volume, quality and geographical coverage of NSP post-GF withdrawal, it was encouraging that NSP provision in both periods covered an array of services recommended by the WHO.[12] Among these, the WHO recommends multiple delivery modalities (NSPs operated both fixed sites and mobile outreach) and referral for first aid, drug treatment, voluntary HIV testing and treatment, diagnosis and treatment of sexually transmitted infections. While several of these services are available to NSP clients, evidence-based drug treatment programmes, such as opioid substitution therapy, have not been scaled up sufficiently in Tijuana.[17] We note that WHO recommends that harm reduction kits include needles and syringes, condoms, filters, sterile water, swabs, spoons, puncture-proof containers, acidifiers,

tourniquets, bleach and other disinfectants and education material. Many of these items were provided during the GF era, and although this provision was drastically reduced post-GF withdrawal, the kits still contained sterile water, alcohol swabs and condoms. Future provision should emphasise increasing coverage of both needles and syringes as well as ancillary kit components for maximum prevention benefit.

While the longitudinal PWID cohort provided some external validation of our findings on diminished NSP provision after GF withdrawal, some caution is warranted when attempting to triangulate these results. Participants in the study did not specify from which NSP (nor from which geographical site) they received syringes; thus, we cannot conclude with any certainty that the participants received their syringes from the NSP that we analysed during and post-GF. Despite this, we still found a highly significant reduced log odds of NSP utilisation in the post-GF period. Overall, this finding is consistent with national-level data from CENSIDA, which reported an 80% decrease in the number of syringes distributed per PWID from 2013 (19.7 syringes per PWID) to 2014 (3.9 syringes per PWID; see online supplementary figure S2).[18]

To our knowledge, our study is the first to estimate the cost of NSP provision in Latin America. Our reported cost per syringe distributed in Tijuana after the GF withdrew support ($0.76 per syringe distributed) is less than estimates in other high-income countries, but higher than other LMIC settings. For example, an analysis in the USA and Canada estimated a cost of $2.28 per syringe distributed, after inflating to 2017 US dollars.[19] In the UK, cost per syringe/needle distributed ranged from $0.28 to $2.17[20] with similar cost per syringe distributed in Australia ($1.94).[21] These differences may be attributed to higher personnel costs which represented 66% of the costs in USA and Canadian NSPs, while this comprised approximately 30%–40% in Tijuana in the post-GF period. Conversely, the cost per harm reduction contact in Mexico was higher than in other LMICs, including Bangladesh ($0.42 in 2017 USD),[7] China ($0.13 in 2017 USD)[22] and Russia ($0.38 in 2017 USD).[23] Despite a similar number of syringes distributed per month, personnel costs were approximately 10 times lower in China compared with Mexico in the post-GF period. However, we cannot determine whether this was due to greater efficiency or fewer personnel.

### Limitations

Our study has several limitations. First, we did not collect any client cost data so we cannot provide any non-service delivery costs, such as PWID transportation to the NSP site. This distinction may be relevant when comparing 2012 with 2015 since PWID were provided more syringes per harm reduction kit in 2012 and therefore may have not needed to access the NSP as frequently (we saw a correspondingly lower number of contacts per month during this period compared with post-GF). In addition, the geographic reach post-GF was much more limited, potentially substantially increasing transport costs for PWID.

We are also uncertain about the number of unique PWID who accessed the NSP during the GF period since we only obtained the number of contacts (kits distributed).

Second, we report on changes in NSP provision during and post-GF support from one NSP in Tijuana; however, other providers (one additional NSP and pharmacies) also provide sterile syringes.[24] However, interviews with the other NSP provider indicate that the reduction in services is likely generalisable. It is legal to obtain syringes from pharmacies; however, PWID often report discrimination and pharmacists' refusal to sell syringes.[25] It is possible that reductions in syringe provision seen within our NSP site due to withdrawal of GF support could have been countered by increased provision of sterile syringes from pharmacies; however, self-reported data from the longitudinal cohort of PWID do not indicate that this occurred (data not shown). Unfortunately, the El Cuete IV survey only began collecting information on number of sterile syringes obtained from any source (NSP, pharmacy, etc) during the last month of GF funding (December 2013) and therefore we were unable to assess individual-level trends before and after. We recognise the importance of collecting these data because previous findings indicate that individual-level coverage affects HIV[26] and HCV risk,[2 27] and these estimates could provide additional insight on monitoring temporal trends in NSP access and provision. Surveillance of clean syringe provision is critical since NSPs are often supported by donors, which could abruptly withdraw funding.

## CONCLUSIONS

In Tijuana, Mexico, abrupt withdrawal of GF support was associated with dramatic declines in coverage and availability of needle and syringe provision, an effective[28] and cost-effective HIV prevention intervention.[29] NSP provision during the GF period in Tijuana involved greater access to sterile syringes at a similar cost per syringe distributed while also providing abundant ancillary items that promoted safe injecting and sexual practices. The withdrawal of multilateral donors has undoubtedly left an impact on other LMIC and is not unique to Mexico. For example Romania, which lost NSP funding from the GF after joining the European Union, saw a precipitous rise in new HIV infections shortly thereafter.[30] In terms of policy implications, it is becoming increasingly important for donors and governments to understand the financial and programmatic implications of abrupt donor withdrawal, and to appropriately plan for transition strategies which ensure that these services are not compromised in terms of quality, coverage and monetary value.

**Author affiliations**
[1]Division of Infectious Diseases and Global Public Health, University of California San Diego, La Jolla, California, USA
[2]Institute for Health Policy Studies, School of Medicine, University of California, San Francisco, California, USA
[3]Centro de Servicios (SER), Tijuana, Mexico
[4]Department of Social and Human Development, State Government of Oaxaca, Oaxaca, Mexico
[5]Prevencasa AC, Tijuana, Mexico
[6]Population Studies, El Colegio de la Frontera Norte, Tijuana, Mexico
[7]National Center for the Prevention and Control of HIV and AIDS, Mexico City, Mexico
[8]School of Social and Community Medicine, University of Bristol, Bristol, UK

**Acknowledgements** We thank the staff from the NSP and all respondents from the El Cuete IV study for participating.

**Contributors** JAC, JLB, SAS and NKM conceived and designed the analysis and contributed to the writing of the manuscript. JAC and JLB conducted the analyses with input from JGK, DA, TG and PV. JLB and PEMM collected cost data with assistance from RP, LAS, GR and CM-R. All authors critically reviewed and approved the final version.

**Funding** This study was funded by grants from the National Institute on Drug Abuse (R01DA037773, K01DA043421, T32DA023356, R01DA015612), Fogarty International Center (R25TW009343) and by the University of California, San Diego, Center for AIDS Research (CFAR), an NIH-funded program (P30AI036214), which is supported by the following NIH Institutes and Centers: NIAID, NCI, NIMH, NIDA, NICHD, NHLBI, NIA, NIGMS and NIDDK.

**Competing interests** NKM has received unrestricted research grants from Gilead unrelated to this work and honoraria from Gilead and Merck.

**Patient consent for publication** Not required.

**Ethics approval** Study procedures were approved by the institutional review boards at University of California San Diego Human Research Protections Program and Universidad Autónoma de Baja California and all NSP personnel provided written informed consent. The study protocol for the El Cuete IV was approved by the University of California San Diego Human Research Protections Program and El Colegio de la Frontera Norte (Tijuana).

**Provenance and peer review** Not commissioned; externally peer reviewed.

**Data sharing statement** Costing spreadsheets are available upon reasonable request. El Cuete IV data cannot be shared due to the need to ensure the confidentiality of participants.

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
