## [Reviewer comments · BMJ Open]

ARTICLE DETAILS

TITLE (PROVISIONAL)	Evaluating the impact of Global Fund withdrawal on needle and syringe provision, cost, and use among people who inject drugs in Tijuana, Mexico: a costing analysis
AUTHORS	Padilla, Rosario; Meza, Emilio; Segovia, Luis Alberto; Gaines, Tommi; Abramovitz, Daniela; Rangel, Gudelia; Magis, Carlos; Vickerman, Peter; Strathdee, Steffanie; Martin, Natasha

VERSION 1 – REVIEW

REVIEWER	V. Anna Gyarmathy Johns Hopkins University, USA
REVIEW RETURNED	20-Sep-2018

GENERAL COMMENTS	This is an interesting manuscript. I have only minor comments. Introduction: The statement that the border traffic resulted in the HIV epidemic is strong: it needs to be supported by at least one reference. Methods: The authors state that there was no patient involvement, even though they say they used data from 734 study participants. It is also unclear from the results section how the 734 interviews were used. This contradiction needs to be resolved. Figures: the font size in the figures is way too small to be legible. Figure 2 looks pretty cute but it is unclear what half a drop of sterile water and a third of a condom really mean. BTW, it took me some time to realize that it was half a condom and not a whip of some sort. The cute drawings need to be labelled for better understanding of the figure by itself, without supporting information from the body text itself. Figure S1: the light and dark blue colors need to be labelled in the figure with a figure legend.
---

REVIEWER	Paul Dietze Burnet Institute, Australia
REVIEW RETURNED	28-Sep-2018

GENERAL COMMENTS	A very interesting paper that effectively provides a pre-post evaluation of a policy/practice change around needle and syringe program provision in Tijuana. The focus is on analysis of health economics parameters and effect with clear indications of negative effects of the withdrawal of Global Fund support for needle and syringe programs in Tijuana. These findings are extremely important as service delivery to marginalised and vulnerable populations is often neglected as the development status of countries improves. I preface my review by pointing out that I am not a health economist and suggest that review by a health economist is warranted. At first the inclusion of the findings from the cohort study seemed somewhat peripheral to the main aims of the paper – it isn't until the discussion that the rationale around triangulation is provided. It would be better if this element were more clearly introduced. However, it is this element of the paper that I think could be enhanced – there are statements about population-level syringe coverage changes noted by CENSIDA which decreased as a result of the withdrawal of the Global Fund support. Nevertheless, it would be of considerable interest to see what changes there were in individual-level syringe coverage (using Bluthenthal et al.'s measure or similar). Given that the authors have access to the cohort study data this information should be reasonably easy to calculate longitudinally (notwithstanding the issue of representativeness acknowledged by the authors) and would strengthen the paper considerably - the provision of 60 syringes per contact in the Global Fund era is a relatively large number, the impact of policy change on individual-level coverage should be assessed. I think some more reflection on the increased number of contacts in the post global fund period is warranted. There is some note of the possible effects here in the limitations section but the impact here seems pretty clear with the increase likely reflecting the reduction in amount of equipment per contact (this further highlights the important of individual-level syringe coverage calculation). I'm not sure about the terminology 'syringe exchange program' – if syringe provision was not contingent on exchange then this terminology should be changed to needle and syringe program (NSP) and equivalent throughout. Otherwise the paper is very well-written.
--

REVIEWER	Chang, Hsing-Yi National Health Research Institutes, Taiwan
REVIEW RETURNED	22-Oct-2018

GENERAL COMMENTS	This study compared the effectiveness of syringe exchange program before and after the support of Global Fund. Authors collected data from Tijuana, Mexico for two periods, in 2012 and 2015-2016. Results showed dramatic reduction in number of contacts per month and non-personnel costs i.e. syringes distributed.  1. Was information in Figure 3 not presented in Table 2? It seemed figure 3 was redundant. 2. What was the policy implication to Mexican government?
--

VERSION 1 – AUTHOR RESPONSE

Reviewer 1 overall comment: This is an interesting manuscript. I have only minor comments.

Author reply: We thank the Reviewer for the positive feedback.

Reviewer 1, comment 1: Introduction: 1. The statement that the border traffic resulted in the HIV epidemic is strong: it needs to be supported by at least one reference.

Author reply: We have now included a reference at the end of this sentence (Mehta SR, et al. EBioMedicine, 2015) which used phylogenetic and network analyses to demonstrate a high level of bidirectional transmission occurring among risk groups in the San Diego – Tijuana border region. (Introduction, pg. 3): Tijuana has a prominent Red-Light district and draws in drug and sex tourists primarily from the United States that has resulted in a localized HIV epidemic, with evidence of bidirectional cross-border HIV transmission occurring among risk groups in the San Diego- Tijuana border region.⁹

Reviewer 1, comment 2: Methods: 2. The authors state that there was no patient involvement, even though they say they used data from 734 study participants.

Author reply: We apologize for the confusion. This statement was written to be compliant with the journal's requirement to report how the public and patients were involved in the research study (please see: <https://blogs.bmj.com/bmjopen/2018/03/23/new-requirements-for-patient-and-public-involvement-statements-in-bmj-open/>). We have revised the patient and public involvement statement and have confirmed that the community was indeed involved with the study prior to its implementation. The public and patient involvement statement is now:

(Methods, pg. 7): The El Cuete IV study was presented to community stakeholders and obtained their approval. Study staff, who had formerly used drugs and reflected the community, reviewed and pre-tested the survey.

Reviewer 1, comment 3: 3. It is also unclear from the results section how the 734 interviews were used. This contradiction needs to be resolved.

Author reply: The data from the 734 interviews were used in the analyses presented in the last paragraph of the results section ("Temporal trends in SEP access among PWID"). We have further clarified this by including the following text at the beginning of this paragraph. (Results, pg. 10): Based on data from PWID in the El Cuete IV study..."

Reviewer 1, comment 4. Figures: the font size in the figures is way too small to be legible.

Author reply: We apologize for the small font size. We have now increased the font size on all figures.

Reviewer 1, comment 5. Figure 2 looks pretty cute but it is unclear what half a drop of sterile water and a third of a condom really mean. BTW, it took me some time to realize that it was half a condom and not a whip of some sort. The cute drawings need to be labelled for better understanding of the figure by itself, without supporting information from the body text itself.

Author reply: We apologize for the confusion. We have now included text in the Figure title that explains why only partial depictions of harm reduction kit components are shown. We have added the following text:

(Figure 2, page 18): Harm reduction kit components per sterile syringe distributed during the Global Fund Period and after withdrawal of Global Fund. Note: Some items are only partially depicted since more syringes per item were distributed (e.g. during the GF period, three syringes were distributed per one condom).

Reviewer 1, comment 6. Figure S1: the light and dark blue colors need to be labelled in the figure with a figure legend.

Author reply: We have now included the figure legend and the following text in the figure title. (Figure S1, page 4): Mean predicted probability of accessing a syringe exchange program in past six months among PWID in Tijuana. Error bars represent 95% confidence intervals. Shaded box represents the calendar period when the Global Fund operated in Mexico.

Reviewer: 2

Reviewer Name: Paul Dietze

Institution and Country: Burnet Institute, Australia

Reviewer 2, comment 1: A very interesting paper that effectively provides a pre-post evaluation of a policy/practice change around needle and syringe program provision in Tijuana. The focus is on analysis of health economics parameters and effect with clear indications of negative effects of the withdrawal of Global Fund support for needle and syringe programs in Tijuana. These findings are extremely important as service delivery to marginalised and vulnerable populations is often neglected as the development status of countries improves. I preface my review by pointing out that I am not a health economist and suggest that review by a health economist is warranted.

Author reply: We thank the Reviewer for highlighting the importance of our work and providing constructive feedback.

Reviewer 2, comment 2: At first the inclusion of the findings from the cohort study seemed somewhat peripheral to the main aims of the paper – it isn't until the discussion that the rationale around triangulation is provided. It would be better if this element were more clearly introduced. However, it is this element of the paper that I think could be enhanced – there are statements about population-level syringe coverage changes noted by CENSIDA which decreased as a result of the withdrawal of the Global Fund support. Nevertheless, it would be of considerable interest to see what changes there were in individual-level syringe coverage (using Bluthenthal et al.'s measure or similar). Given that the authors have access to the cohort study data this information should be reasonably easy to calculate longitudinally (notwithstanding the issue of representativeness acknowledged by the authors) and would strengthen the paper considerably - the provision of 60 syringes per contact in the Global Fund era is a relatively large number, the impact of policy change on individual-level coverage should be assessed.

Author reply: We thank the Reviewer for this excellent recommendation. Unfortunately, the El Cuete IV survey only began collecting information on number of sterile syringes obtained from any source (NSP, pharmacy, etc.) during the last month of Global Fund funding (December 2013) and therefore we were unable to assess individual-level trends before and after. Nonetheless, from the survey period where the number of sterile syringes was asked, we adapted Bluthenthal's calculation by dividing the number of clean syringes obtained per month by the total number of injections per month to obtain the proportion of injections which were done with clean syringes. We grouped the coverage into three levels <50%, 51-99%, 100% (as done by Bluthenthal). Overall, the proportion of high syringe coverage (>100%) was low (<10%) and did not significantly change from December 2013 onwards. Additionally, El Cuete IV did not elicit information on the number of sterile syringes obtained from specific providers (e.g. NSP vs pharmacy), and therefore we are unable to estimate changes in acquisition from specific sources. Unfortunately, without data from the GF period, it is difficult to determine whether there was already a declining trend in the coverage of clean syringes, and we cannot determine whether any potential declines in coverage due to NSP provision were countered by increased provision by other sources.

We have noted in the limitations the importance of capturing individual level clean syringe coverage, which could be useful to monitor temporal changes in clean syringe access and provision. We have noted this in the limitations section:

(Limitations, page 13): Unfortunately, the El Cuete IV survey only began collecting information on number of sterile syringes obtained from any source (NSP, pharmacy, etc.) during the last month of Global Fund funding (December 2013) and therefore we were unable to assess individual-level trends before and after. We recognize the importance of collecting these data because previous findings indicate that individual level coverage affects HIV (Bluthenthal et al., Drug Alcohol Depen, 2007) and HCV risk (Platt et al., Cochrane database of systematic reviews, 2017; Turner et al., Addiction, 2011), and these estimates could provide additional insight on monitoring temporal trends in NSP access and provision. Surveillance of clean syringe provision is critical since NSPs are often supported by donors, which could abruptly withdraw funding.

Reviewer 2, comment 3: I think some more reflection on the increased number of contacts in the post global fund period is warranted. There is some note of the possible effects here in the limitations section but the impact here seems pretty clear with the increase likely reflecting the reduction in amount of equipment per contact (this further highlights the important of individual-level syringe coverage calculation).

Author reply: Please see our previous response as we have now added the importance of individual level syringe coverage in the limitations section.

Reviewer 2, comment 4: I'm not sure about the terminology 'syringe exchange program' – if syringe provision was not contingent on exchange then this terminology should be changed to needle and syringe program (NSP) and equivalent throughout. Otherwise the paper is very well-written.

Author reply: We thank the Reviewer for this clarification. As syringe provision was not contingent on exchange at this site in Tijuana, we have changed SEP to NSP throughout the manuscript.

Reviewer: 3

Reviewer Name: Hsing-Yi Chang

Institution and Country: National Health Research Institutes, Taiwan

Reviewer 3, comment 1: This study compared the effectiveness of syringe exchange program before and after the support of Global Fund. Authors collected data from Tijuana, Mexico for two periods, in 2012 and 2015-2016. Results showed dramatic reduction in number of contacts per month and non-personnel costs i.e. syringes distributed.

Reviewer 3, comment 2. Was information in Figure 3 not presented in Table 2? It seemed figure 3 was redundant.

Author reply: While much of the data presented in Table 2 can be found in Figure 3, we believe that the visual depiction of the cost breakdown is important for the reader. Specifically, it is important to highlight that while the cost per syringe distributed did not decrease, the distribution of how the cost per syringe did change substantially post Global Fund. Additionally, we also wished to highlight that in order to adjust for reduced costs, many of the ancillary kit components were no longer distributed in the post GF-period which is more striking in the figure than in the table.

Reviewer 3, comment 3. What was the policy implication to Mexican government?

Author reply: We thank the Reviewer for posing this important question. The policy implication to the Mexican government is to ensure that upon donor withdrawal (as the Global Fund did in 2014), a

transitional plan should be implemented so that the government can provide funding without loss of provision of services. Based on the available data, we did not see equivalent levels of syringe provision after the Global Fund withdrew in 2014. We have clarified this in the last paragraph of the conclusion section as detailed below. Additionally, we wish to note that this situation of abrupt donor withdrawal to fund needle and syringe programs is not unique to Mexico and has been observed in other countries with high prevalence of injection drug use.

(Conclusion, page 14): The withdrawal of multilateral donors has undoubtedly left an impact on other LMIC and is not unique to Mexico. For example, Romania, which lost NSP funding from the Global Fund after joining the EU, saw a precipitous rise in new HIV infections shortly thereafter. In terms of policy implications, it is becoming increasingly important for donors and governments to understand the financial and programmatic implications of abrupt donor withdrawal, and to appropriately plan for transition strategies which ensure that these services are not compromised in terms of quality, coverage, and monetary value.